# Symmetry-breaking dynamics in a tautomeric 3D covalent organic framework

Yangyang Xu[1,4], Tu Sun[1,2,4], Tengwu Zeng[1,4], Xiangyu Zhang[1], Xuan Yao[1], Shan Liu[1], Zhaolin Shi[1], Wen Wen[3], Yingbo Zhao [1,2], Shan Jiang[1,2], Yanhang Ma [1,2] & Yue-Biao Zhang [1,2] ✉

The enolimine-ketoenamine tautomerism has been utilised to construct 2D covalent organic frameworks (COFs) with a higher level of chemical robustness and superior photoelectronic activity. However, it remains challenging to fully control the tautomeric states and correlate their tautomeric structure-photoelectronic properties due to the mobile equilibrium of proton transfer between two other atoms. We show that symmetry-asymmetry tautomerisation from diiminol to iminol/*cis*-ketoenamine can be stabilised and switched in a crystalline, porous, and dynamic 3D COF (dynaCOF-301) through concerted structural transformation and host-guest interactions upon removal and adaptive inclusion of various guest molecules. Specifically, the tautomeric dynaCOF-301 is constructed by linking the hydroquinone with a tetrahedral building block through imine linkages to form 7-fold interwoven diamondoid networks with 1D channels. Reversible framework deformation and ordering-disordering transition are determined from solvated to activated and hydrated phases, accompanied by solvatochromic and hydrochromic effects useful for rapid, steady, and visual naked-eye chemosensing.

Prototropic tautomerism is one of the most important phenomena in physical, organic, supramolecular, materials, and biological chemistry, featuring mobile equilibrium of intramolecular proton transfer and environmentally sensitive photoelectronic absorption/emission spectra[1–5]. However, the small free energy differences and the low energy barrier between the tautomers impose great challenges to their isolation, characterisation, and correlation of the tautomeric structure-photoelectronic properties[1]. Soft porous crystals (SPCs) are an ideal platform for studying prototropic tautomerism, featuring global crystal structural transformation propagated from the local conformational/configurational responses to external stimuli leading to the amplification and multiplication of output signals[6–13]. Dynamic covalent organic frameworks (dynaCOFs) are an emerging form of SPCs constructed by stitching organic building blocks into extended networks through strong covalent bonds[14–22], featuring concerted

framework deformation and multiplex electronic structure transitions for guest-adaptive molecular sensing[23–31]. Recently, enolimine-ketoenamine tautomerism has endowed remarkable improvements in crystallinity, porosity, chemical stability, and photoelectronic activities in 2D COFs[32–38]. The tautomerism from diiminol to iminol/*cis*-ketoenamine has been implemented in a 2D COF for humidity sensing with visible colour change[38]. However, their local structure changes could hardly be observed due to insufficient crystallinity and subtle structural change.

Herein, we report the symmetry-breaking dynamics by the symmetry-asymmetry tautomerisation from diiminol to iminol/*cis*-ketoenamine in a solvatochromic and hydrochromic dynaCOF for rapid, steady, and visual naked-eye humidity sensing (Fig. 1). Specifically, the tautomeric dynaCOF-301 was constructed by stitching the hydroquinone with a tetrahedral building block through imine

[1]School of Physical Science and Technology, ShanghaiTech University, Shanghai 201210, China. [2]Shanghai Key Laboratory of High-Resolution Electron Microscopy, ShanghaiTech University, Shanghai 201210, China. [3]Shanghai Synchrotron Radiation Facility, Shanghai Advanced Research Institute, Chinese Academic of Sciences, Shanghai 201210, China. [4]These authors contributed equally: Yangyang Xu, Tu Sun, Tengwu Zeng. ✉e-mail: zhangyb@shanghaitech.edu.cn

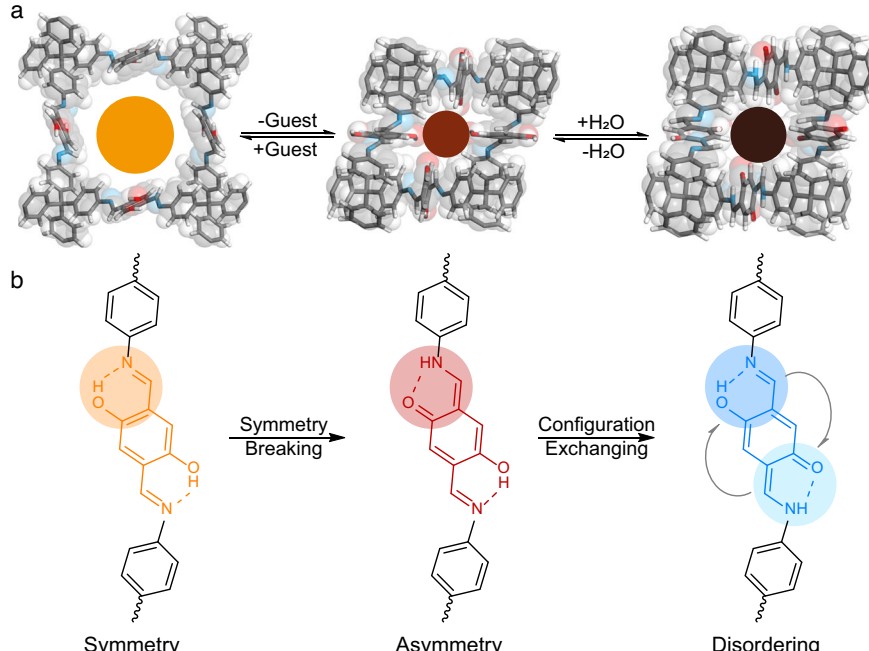

**Fig. 1 | The symmetry-breaking dynamics in the tautomeric 3D COF. a** De-symmetry transformation of dynaCOF-301 from solvated (crystallised in tetragonal $I4_1/a$, sample colour: orange) to activated (crystallised in monoclinic $I2/a$, sample colour: red) and hydrated (crystallised in monoclinic $I2/a$, sample colour: dark brown) phases, exhibiting solvatochromic and hydrochromic effects. **b** Local chemical tautomerisation from symmetric diiminol (solvated) to asymmetric iminol/cis-ketoenamine (activated) resulting in compromised crystallinity and rapid exchange of configurations between iminol/cis-ketoenamine and cis-ketoenamine/iminol to achieve long-range ordering.

linkages to from 7-fold interwoven diamondoid networks with 1D channels. De-symmetry crystal structural transformation from a solvated (tetragonal $I4_1/a$, no. 88) to an activated phase (monoclinic $I2/a$, no. 15) was determined by 3D electron diffraction (3D ED) and synchrotron powder X-ray diffraction (PXRD) analyses, representing a large amplitude of crystal structural contraction up to ~63 vol.%. Reversible framework deformation was uncovered by in-situ PXRD during gas adsorption, undergoing framework deformation recovery and expansion. A moisture-induced local ordering-disordering transition while retaining the overall crystal symmetry led to a crystallinity enhancement unravelled by in-situ PXRD during water vapour adsorption. This de-symmetry transformation is driven by the diiminol-iminol/cis-ketoenamine tautomerism upon guest removal, and the ordering-disordering is attributed to water-assisted rapid isomerisation upon hydration. The diiminol-iminol/cis-ketoenamine tautomerism is evident by 3D ED, solid-state nuclear magnetic resonance (ssNMR), and diffuse reflectance spectroscopy (DRS). The resulting hydrochromic effects were quantitatively studied by static and dynamic water vapour adsorption and DRS showing great potential in rapid, steady, and visible naked-eye humidity sensing. These results shed light on the design of functional porous materials with deliberate control of external-stimuli responses by integrating crystal dynamics with tautomeric structure and photoelectrical properties.

## Results

### Preparation and characterisation of tautomeric dynaCOF-301
The microcrystals of dynaCOF-301 were prepared from imine condensation of tetra-(4-anilyl)-methane (TAM) and 2,5-dihydroxyterephthalaldehyde (dhTPA) according to our established ventilation-vial synthetic protocol[24] (Fig. 2a; Supplementary Section 1). Considering the size-dependent dynamics[27], we added aniline as a modulating agent in the synthesis to control the crystal size distribution. Consequently, uniform morphology in tetragonal-prismatic shapes was shown by scanning electron

microscopy (SEM, Fig. 2b) with a relatively narrow size distribution centred at 3 μm (Fig. 2c). The Fourier-transformed infra-red spectroscopy (FT-IR, Fig. 2d) shows the formation of imine bonds (i.e., −C=N− stretching at 1611 cm$^{-1}$) and the elimination of starting materials (i.e., without −C=O stretching at 1653 cm$^{-1}$). Thermal gravimetric analysis (TGA, Supplementary Fig. 1) indicates thermal stability up to 400 °C, comparable with dynaCOF-300. The chemical stability was highlighted by immersing the sample in 2 M NaOH, retaining its crystallinity for at least 7 days (Supplementary Fig. 2). Comparison of the PXRD patterns for dynaCOF-301 with starting materials shows very different patterns (Fig. 2e), excluding the possibility of recrystallisation of the starting materials. PXRD patterns for the solvated (dynaCOF-301s), activated (dynaCOF-301a), and hydrated (dynaCOF-301h) samples all exhibit sharp peaks but with distinct patterns (Fig. 2e), indicating the possibility of dynamic structure transformation.

### Dynamic texture characterisation by gas adsorption isotherms
The gas adsorption isotherms were collected using various gases as probes at each critical temperature to examine the dynamic texture of dynaCOF-301. The dynaCOF-301 represents different dynamic responses from those of the dynaCOF-300 illustrated here by the adsorption isotherms for $N_2$ at 77 K (Fig. 2f), $CH_4$ at 112 K (Fig. 2g) and $CO_2$ at 195 K (Fig. 2h). A significantly higher $N_2$ uptake (350 cm$^3$ g$^{-1}$) and a larger pore volume (0.57 cm$^3$ g$^{-1}$) were observed for dynaCOF-301 (Fig. 2f) than those of the dynaCOF-300 (only ~200 cm$^3$ g$^{-1}$ and 0.32 cm$^3$ g$^{-1}$), suggesting a full crystal expansion for dynaCOF-301 but only partial conversion for dynaCOF-300. Unexpectedly, a three-step adsorption isotherm of $N_2$ is observed with a steep uptake at $P/P_0$-0.001, a second uptake at $P/P_0$-0.05, and a shallow uptake at $P/P_0$-0.4 for dynaCOF-301. Lower $CH_4$ and $CO_2$ uptakes and smaller pore volumes (0.46 and 0.33 cm$^3$ g$^{-1}$) were observed (Figs. 2g and 2h). Interestingly, the third step uptake of $CO_2$ might be attributed to the re-arrangement of $CO_2$ adsorbates at high $P/P_0$-0.8. Overall, such

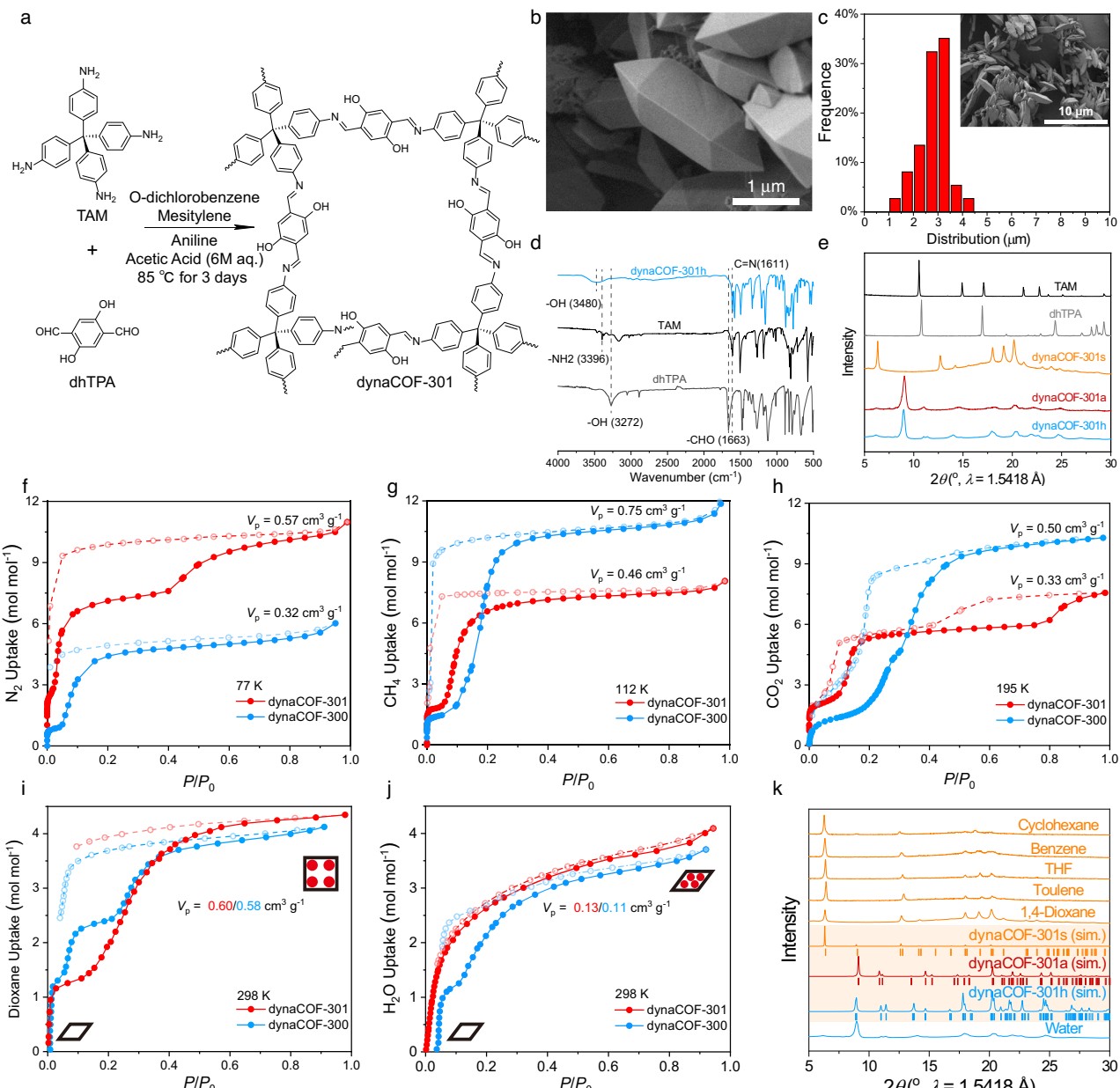

**Fig. 2 | Preparation and characterisation of dynaCOF-301. a** Facile synthetic protocol of high-quality microcrystals with size control. **b** Well-defined morphology of the tetragonal-prismatic microcrystals (scale bar: 1 μm). **c** Statistic histogram of the crystal size distribution centred at 3 μm measured from the SEM image (inset). **d** Comparison of IR spectra of the hydrated sample with its starting materials. **e** Comparison of PXRD patterns of starting materials with the solvated, activated, and hydrated samples. **f** $N_2$ adsorption isotherms at 77 K. **g** $CH_4$ adsorption isotherms at 112 K. **h** $CO_2$ adsorption isotherm at 195 K. **i** 1,4-Dioxane adsorption isotherm at 298 K. **j** $H_2O$ adsorption isotherm at 298 K for dynaCOF-301a (red) compared with dynaCOF-300 (blue). **k** Comparison of PXRD patterns for dynaCOF-301a exposed to various organic vapours and water.

dynamic responses to various gas molecules at lower temperatures are of great interest for understanding their structural evolution.

## Dynamic texture characterisation by vapour adsorption isotherms

The porosity of dynaCOF-301 can also be assessed by vapour adsorption of 1,4-dioxane at 298 K (Fig. 2i), exhibiting two-step adsorption and hysteresis desorption reaching a pore volume of 0.58 cm³ g⁻¹ consistent with that calculated from the $N_2$ adsorption isotherm. As a comparison, dynaCOF-300 displays three-step adsorption to have a slightly higher pore volume of 0.60 cm³ g⁻[124]. $H_2O$ adsorption isotherm at 298 K (Fig. 2j) shows steep one-step uptake at a much lower relative pressure ($P/P_0$-0.01) and with a knee shape isotherm with shallow

uptake starting from $P/P_0$-0.2, which is different from that of the dynaCOF-300 having a two-step water uptake due to the $H_2O$ induced the crystal contraction. This result shows that dynaCOF-301 retained its deformed phase upon water adsorption to afford a small pore volume of 0.11 cm³ g⁻¹ comparable with dynaCOF-300. Comparison of PXRD patterns (Fig. 2k) exposed to various organic vapours and moisture, such as cyclohexane, benzene, toluene, acetone, and tetrahydrofuran (THF), support the generality of crystal expansion of dynaCOF-301 in other organic solvents.

## Atomic-resolution single-crystal structures by 3D ED analyses

The single-crystal structures of dynaCOF-301 were determined by 3D ED with a cryo-holder (Fig. 3a–e; Supplementary Section 2)[25, 26, 39, 40].

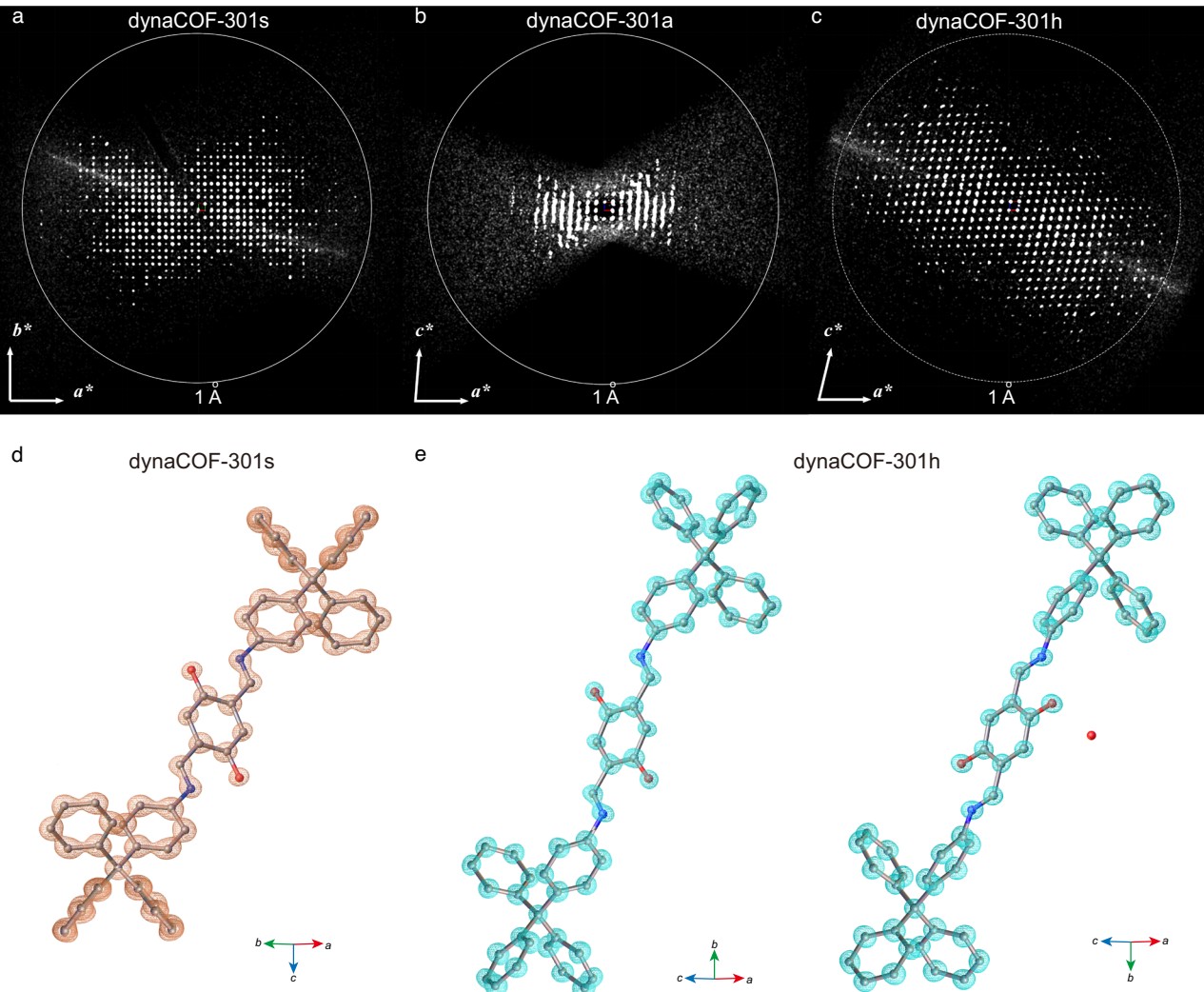

**Fig. 3 | Atomic-resolution single-crystal structures of dynaCOF-301 determined by 3D cryo-ED. a** Reconstructed ED data for dynaCOF-301s projected along the [001] direction with resolution up to ~1.0 Å. **b** Reconstructed ED data for dynaCOF-301a projected along the [010] direction with limited resolution. **c** Reconstructed ED data for dynaCOF-301h projected along the [010] direction with resolution up better than 1.0 Å. **d** The asymmetric unit of the crystal structure with the electrostatic potential density of dynaCOF-301s solved in tetragonal $I4_1/a$ space group. **e** The asymmetric units of the crystal structure with the electrostatic potential density of dynaCOF-301s solved in tetragonal $I2/a$ space group.

The expanded phase crystallises in the tetragonal space group $I4_1/a$ (No. 88) with lattice constants of $a = 26.75$ Å, $c = 7.38$ Å, and $V = 5276$ Å$^3$ with ED data resolution up to 1.0 Å and completeness of 87.3% (Fig. 3a; Supplementary Table 1), enabling the ab initio structure solution using direct methods (Fig. 3d; Supplementary Fig. 3). The residual electrostatic potential density indicates the presence of proton bonding to the O atom with the apparent O·H distance of 1.340 Å from the residual electrostatic density map (Supplementary Fig. 11). The activated phases give only low-resolution data (Fig. 3b; Supplementary Fig. 4) with/without cryo-holder. Despite the poor resolution, the unit cell parameters could be determined as $a = 20.80$ Å, $b = 8.90$ Å, $c = 21.10$ Å, $\beta = 95.0°$, and $V = 3891$ Å$^3$ crystallising in monoclinic space groups $I2/a$ (no. 15). The hydrated phase crystallises were determined in the monoclinic space group of $I2/a$ (no. 15) with unit cell parameters of $a = 20.07$ Å, $b = 8.81$ Å, $c = 20.25$ Å, $\beta = 100.5°$, and $V = 3520$ Å$^3$, having resolution up to 1.0 Å and completeness of 97.4% after merging multiple datasets (Fig. 3c; Supplementary Figs. 5 and 8). Due to lower symmetry, the asymmetric unit possesses two independent positions of the proton between the O and N (Fig. 3e; Supplementary Fig. 10), with O−H distances of 1.368 and 1.490 Å and N−H distances of 1.390 and 1.068 Å, respectively, from the residual electrostatic potential density map (Supplementary Fig. 12). These results reveal the prototropic tautomerisation from ordered diiminol in dynaCOF-301s to disordered iminol/*cis*-ketoenamine in dynaCOF-301h.

### Bulk-sample crystal structures by synchrotron PXRD analyses

The solvated, activated, and hydrated crystal structures for bulk samples were determined by synchrotron PXRD with Rietveld refinements (Fig. 4; Supplementary Tables 2–5). Remarkably, de-symmetry transformations were observed from solvated ($I4_1/a$, no. 88) to activated/hydrated phases ($I2/a$, no. 15), with the unit cell volume shrinking by ~62.8% (Fig. 4a). A disordered to ordered transition is indicated by full widths at half maximum (FWHM) of the 200/002 reflections from 0.104° to 0.081° (Supplementary Fig. 13). The square channel for the solvated phase deformed into a diamond shape after activation, which retained the shape upon hydration, with the pore diameter contracting from 9 Å to 2.8 and 3.4 Å for dynaCOF-301a and dynaCOF-301h, respectively (Fig. 1a). Moreover, the positions of guest molecules were successfully located, revealing hydrogen bonding formations between guest molecules $H_2O$ and the framework (Supplementary Fig. 16). The mechanism of how dynaCOF-301 adapted itself upon

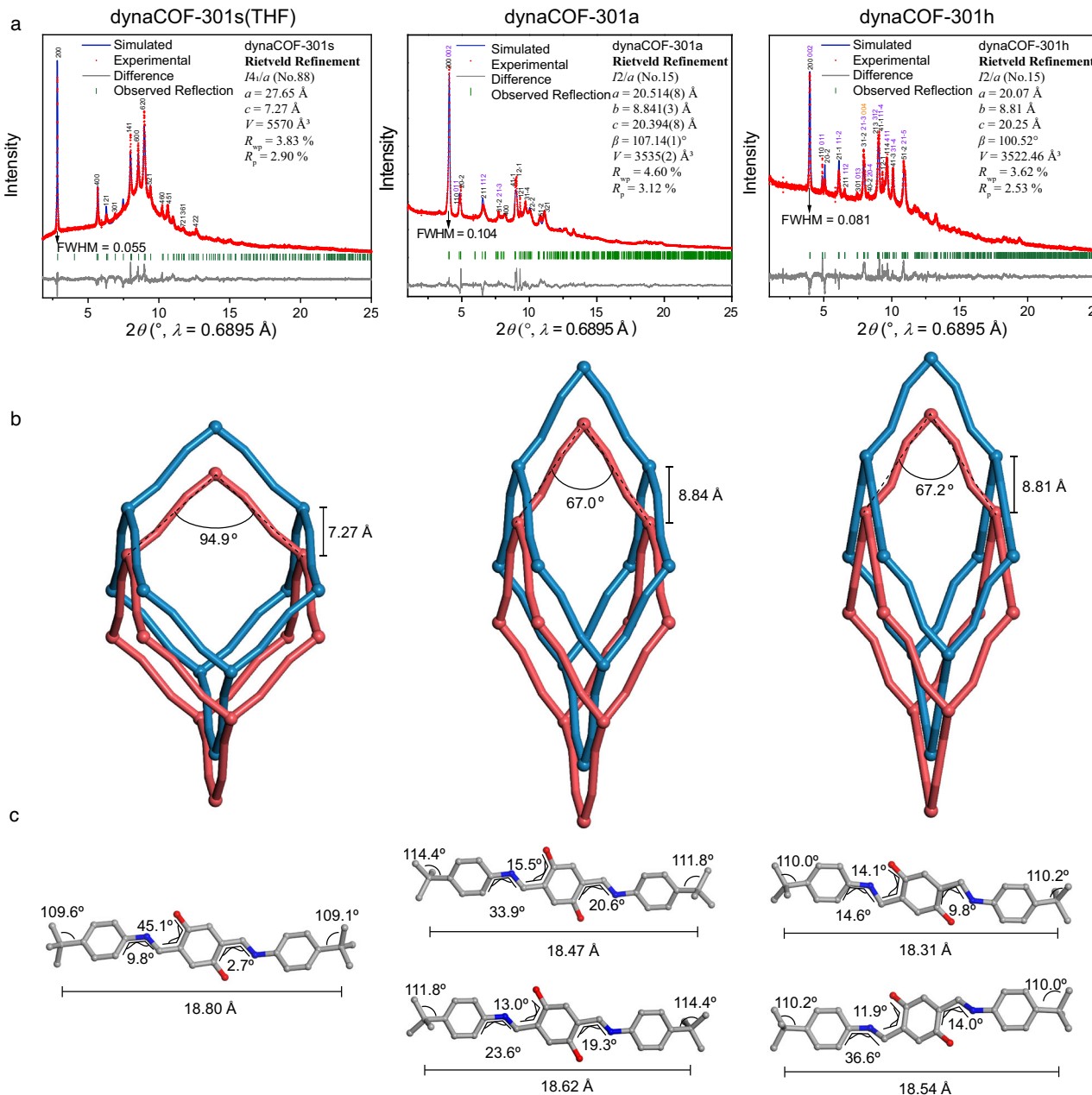

**Fig. 4 | De-symmetry transformation of the bulk sample determined by synchrotron PXRD analyses. a** Indexed synchrotron PXRD patterns (red) and refined profiles (blue) using the Rietveld method along with the Bragg reflections (green bar) and difference (grey) for solvated (THF), activated, and hydrated bulk samples. **b** Variation of framework geometry and displacement between frameworks. **c** Molecular geometries and conformational changes in dynaCOF-301. (Colour codes: C, grey; N, blue; O, red.).

solvation, activation, and hydration is depicted (Fig. 4b, c), attributing to the deformation of node geometry, the displacement between frameworks, and the conformation changes of the organic linkers. The node geometries changing from 94.9° to 67.0/67.2° is responsible for the crystal contractions upon activation/hydration, which is actualised by the organic linkers' conformation changes. Due to the lower symmetries, the asymmetric units of the activated and hydrated phases were doubled, enabling more freedom of conformation changes.

## Structural evolution during gas adsorption

To track the structural transformation during gas adsorption at room temperature, *n*-butane ($C_4H_{10}$) gas adsorption isotherms at 298 K were collected, which exhibited stepwise uptake and hysteresis desorption and approached a pore volume of 0.55 cm³ g⁻¹ (Fig. 5a). Then, in-situ

PXRD patterns were collected under butane gas flow with $N_2$ as the carrier gas in various concentrations (Fig. 5b, c, Supplementary Fig. 18). The PXRD patterns of dynaCOF-301 under $N_2$ and 0.25% butane gas flows maintain the low-symmetry activated phase (Fig. 5b, red). A distinguishable lattice change was observed under 1% butane flow (Fig. 5b, brown), comparable with the contracted phase of dynaCOF-300 in the tetragonal $I4_1/a$ space group. This suggests the deformation recovery of the activated phase of dynaCOF-301 corresponding to a steep uptake at 1 kPa (Fig. 5a). A new reflection peak shows up at 6.6° while the original 200/002 reflection peak intensity at 8.8–8.9° gradually decreases under 5% and 10% butane flow. It is worth noting that an extra peak was also observed, revealing a possible intermediate phase. Lattice expansion completed under >20% butane flow (Fig. 5b, orange), corresponding to the second steep uptake turning at 20 kPa

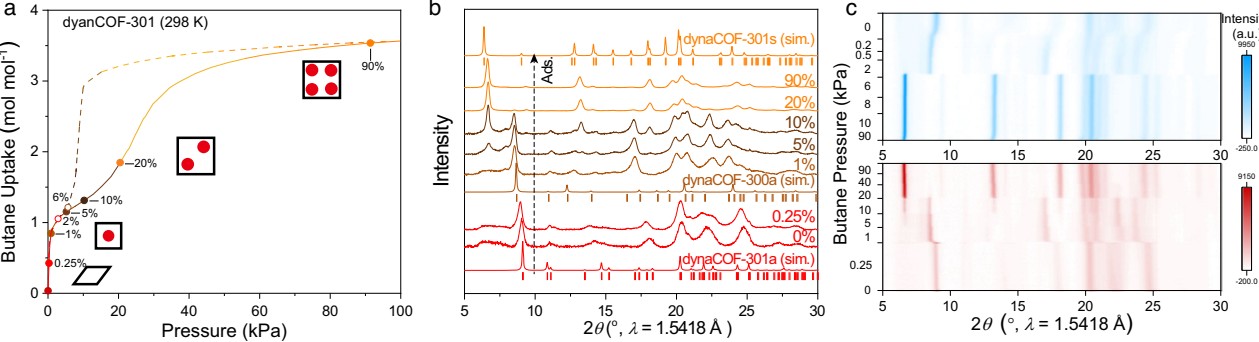

**Fig. 5 | Reversible structural evolution of deformed dynaCOF-301 during gas adsorption undergoes framework deformation, recovery, and expansion.** **a** Butane gas adsorption isotherm at 298 K coloured for different stages. **b** Comparison of the representative in-situ PXRD patterns under varied-concentration butane gas flows with the simulated patterns of dynaCOF-301a, dynaCOF-300a, and dynaCOF-301s. **c** Contour plot of the in-situ PXRD patterns under varied concentrations of butane flows at room temperature (red, adsorption; blue, desorption).

(Fig. 5a). Comparison of the PXRD pattern under 90% butane flow with the simulated pattern of the solvated phase (Fig. 5b) suggests less expansion after the adaptive inclusion of butane flow than that was observed in THF solvent. In-situ PXRD also confirmed the reversible structural transition under butane gas with decreasing concentrations (Fig. 5c; Supplementary Fig. 18). The crystal contraction occurred under 6% butane flow and almost completed the transition under 2%.

To gain energetic insight into the dynamic structural transformation, molecular dynamics (MD) simulations were implemented to statistically model the transition from expanded to contracted phases of dynaCOF-301 (Supplementary Section 7, Supplementary Movie 1)[41]. The internal pressure of dynaCOF-301 as a function of variable cell volumes was then calculated (Supplementary Fig. 32a). Stable structures with zero internal pressure are suggested with cell volumes of 3734 and 5844 Å³, comparable to observed cell volumes 3535 Å³ for dynaCOF-301a and 5570 Å³ for dynaCOF-301s, respectively. Surprisingly, a metastable structure with 5213 Å³ cell volume was suggested as an intermediate phase during the gas adsorption. From the energy profile (Supplementary Fig. 32a), the free energy of the expanded framework is 60 kJ/mol higher than that of the contracted one, whose energy barrier can be conquered by the adaptive inclusion of guest molecules.

**Solvatochromic and hydrochromic effects**

The activated samples were exposed to various organic solvents with polarities, such as hexane, toluene, 1,4-dioxane, THF, and acetonitrile (Supplementary Table 7). The light-brown sample turns to light yellow colours distinguished by the naked eye. In contrast, we observed a significant colour change to dark brown upon exposure to atmospheric moisture. The DRS of the capillary-sealed samples dosed with organic vapours was then collected (Fig. 6a). The bandgaps of the activated sample were analysed[42] as 2.2 and 1.8 eV, corresponding to the iminol/*cis*-ketoenamine configuration. The solvated samples represent only single bandgaps at 2.1–2.2 eV, corresponding to the diiminol configuration. In comparison, the hydrated sample represents only single bandgaps at 1.83 eV, which can be attributed to the rapid configuration exchange between iminol/*cis*-ketoenamine and *cis*-ketoenamine/iminol. With the potential of molecular sensing, absorbance intensity at 600 nm was measured according to the polarity of the included solvent (Fig. 6b). These results reveal that the guest molecules could control the direction of tautomerism, and the water uptake can facilitate the configuration exchange.

**Distinguishable local chemical structures**

To understand the solvatochromic and hydrochromic effects of dynaCOF-301, high-resolution ¹³C ssNMR spectroscopy with cross-polarised/magic angle spinning (CP/MAS) techniques was performed with solvated, activated, and hydrated samples (Supplementary Section 6). Chemical shifts of the quaternary carbons were assigned by varying the value of CP contact time (Supplementary Figs. 28 and 29).

All the characteristic ¹³C chemical shifts were well-resolved. The spectra are distinct for the THF-solvated, activated, and hydrated samples (Fig. 6c). The chemical shift at 160.7 ppm (C6) characteristic for the imine carbon for dynaCOF-301s split into two peaks at 160.2 and 156.4 ppm (C6 and C6′) for dynaCOF-301a, while the chemical shift at 153.0 ppm (C9) assigned to the α-carbon of hydroquinone split into two peaks at 153.5 and 152.5 ppm (C9 and C9′), indicating the tautomerism of diiminol to iminol/*cis*-ketoenamine. Variable-temperature ssNMR further confirmed temperature sensitivity of the enolimine-ketoenamine tautomerism for dynaCOF-301a showing distinguishable intensity changes of chemical shifts of C9 and C9′ at 243–352 K (Supplementary Fig. 30). We also conducted the ex-situ ssNMR experiments by dosing various amounts 1,4-dioxane vapour to dynaCOF-301a for probing the local chemical environment changes of host framework upon adaptive inclusion of guest molecules (Supplementary Fig. 31). Upon 1 mol mol⁻¹ inclusion of 1,4-dioxane, the chemical shifts of C9 and C9′ converged to one chemical shift at 153.0 ppm implying the tautomerism to diiminol form and the gradual recovery of high symmetry. Further inclusion of 1,4-dioxane leads to the chemical shifts of C6 and C6′ converging to one chemical shift at 160.7 ppm, indicating the conformation change of imine bonds.

On the other hand, the C6/C6′ and C9/C9′ chemical shifts in dynaCOF-301h converged at 159 and 153.2/152.4 ppm, respectively, which were attributed to the rapid tautomerisation and aromatic resonance facilitated by $H_2O$ adsorbates. Notably, the FWHM of chemical shifts hints at the crystallographic symmetric and crystallinity of the three states. The C1 chemical shifts at 65.5 ppm are as narrow as 40 Hz, showing high crystallinity. With the same symmetry, the C1 chemical shift at 65.6 ppm for dynaCOF-301a (120 Hz) is broader than that of the dynaCOF-301h (97 Hz), which is consistent with the results of 3D ED and PXRD analyses.

**Rapid, steady, and visual naked-eye humidity sensing**

To explore the potential for humidity sensing of dynaCOF-301, the colour changes during $H_2O$ adsorption were recorded (Fig. 6d), which suggested a significant colour change from $P/P_0$-0.10 to 0.99. DRS of dynaCOF-301 at various $P/P_0$ (Fig. 6e) has shown significantly different spectra for 22–99 %RH, while the one at 1% is distinct from others. At $P/P_0$-0.01 to 0.49, the dynaCOF-301 possesses two bandgaps at 1.80–1.83 and 2.17–2.23 eV, while only one band gap at 1.80–1.83 eV at $P/P_0$-0.68 to 0.99. Therefore, it is possible to exploit such colour changes for humidity sensing as a function of various RH.

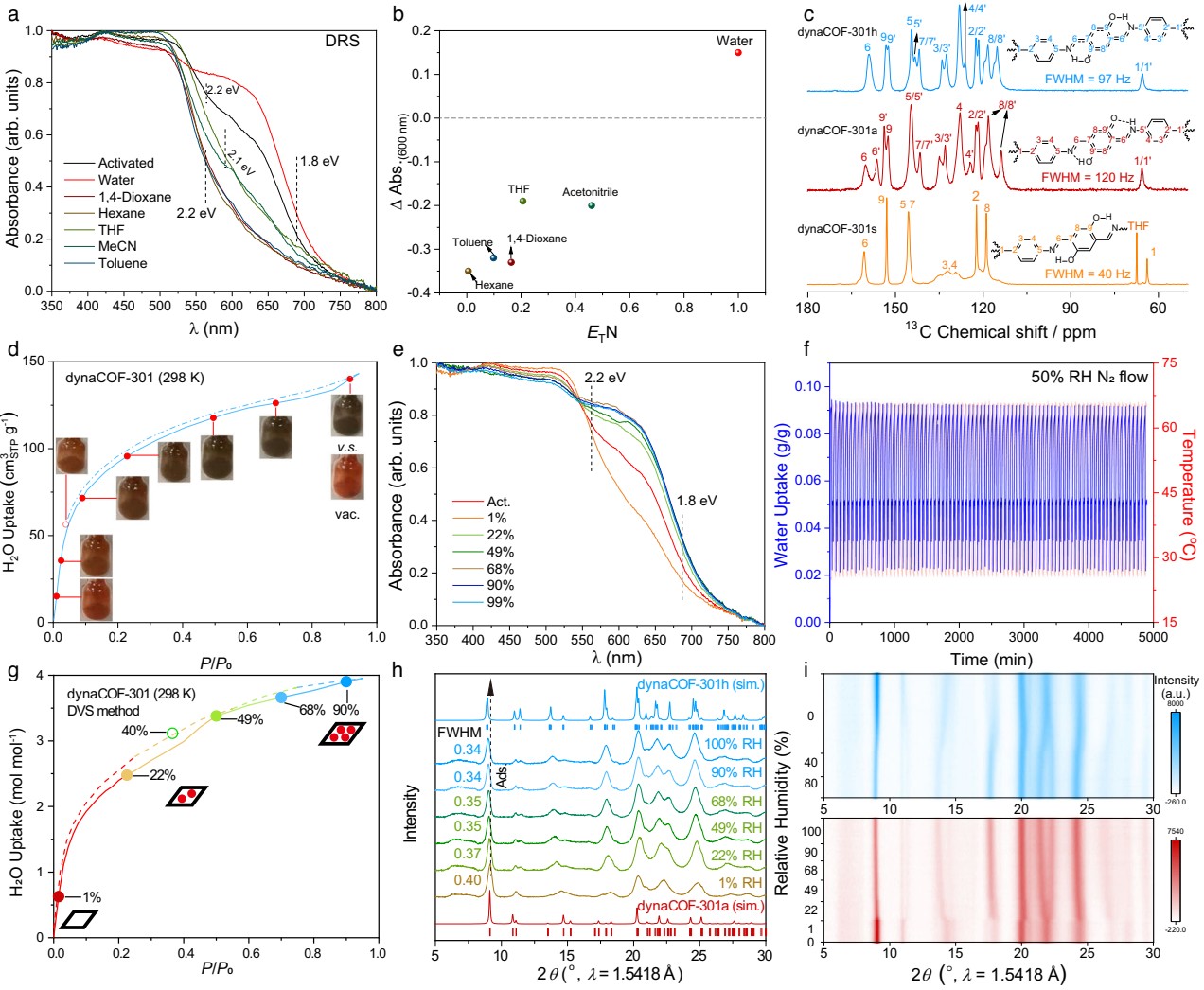

**Fig. 6 | Solvatochromic and hydrochromic effects, diiminol-iminol/*cis*-ketoenamine tautomerisation, and ordering-disordering transition of dynaCOF-301.**
**a** Quantitative spectra of dynaCOF-301a dosed with various organic vapours. **b** The relationship of the absorbance difference at 600 nm to the polarity and exposed solvents. **c** $^{13}$C CP-MAS solid-state NMR spectra with the assignment of chemical shifts indicated in the chemical structure. **d** Obvious colour change of dynaCOF-301a during $H_2O$ adsorption. **e** Quantitative spectra of dynaCOF-301a dosed with different amounts of $H_2O$ and corresponding energy bands. **f** Cyclic water uptakes under 50% RH by swinging temperatures between 25 and 65 °C. **g** Dynamic $H_2O$ vapour adsorption isotherm at 298 K. **h** In-situ PXRD patterns under various RHs $N_2$ gas flows showing the variation of FWHM. **i** Contour plot of the in-situ PXRD patterns (red, adsorption; blue, desorption) under varied RHs $N_2$ flow at room temperature.

Furthermore, the cycling water uptake under 50% RH and 298 K validates that dynaCOF-301 has rapid and steady water uptake that retains more than 90% after 80 cycles (Fig. 6f).

**Disorder-ordering transition upon moisture uptakes**
For tracking the structural transition upon water adsorption, $H_2O/N_2$ dynamic vapour sorption isotherm (DVS, Fig. 6h) and in-situ PXRD patterns of dynaCOF-301 (Fig. 6i) were collected at room temperature and under various relative humidity (RH). Retaining the same crystallographic symmetry, the water uptake of dynaCOF-301 exhibits a one-step Type I isotherm (Fig. 2k). With $N_2$ as the carrier gas, the gravimetric water uptake represents a hollow stepwise uptake (Fig. 6h), which is slightly different from the one based on volumetric method (Supplementary Fig. 19). The in-situ PXRD patterns of dynaCOF-301a under various RH (Fig. 6i) exhibit similar lattice constants. At 1% RH, the PXRD pattern is very close to the activated phase, which switches to the hydrated phase after 22% RH, with the full width at half maximum (FWHM) of 200 reflection peaks narrowing from 0.40° to 0.34–0.37° (Fig. 4f; Supplementary Table 6). During the desorption

process, peaks shift gradually in a reversible manner without hysteresis (Supplementary Fig. 20). These results suggest a crystallinity enhancement upon water uptake by accelerating the configuration exchange for long-range ordering pore walls.

In summary, we have integrated the enolimine-ketoenamine tautomerism with concerted structural transformation in a crystalline, porous, and dynamic hydroquinone-based 3D COF to show symmetry-breaking dynamics. The elusive prototropic tautomerism is stabilised, characterised, and controlled through the crystal structural transformation upon removal and adaptive inclusion of various guest molecules, which is very challenging to study in gas, solution, and dense solids. Particularly, we observed the symmetry-asymmetry tautomerism from diiminol to iminol/*cis*-ketoenamine switched by the removal and adaptive inclusion of guest molecules in dynaCOF-301. We uncovered the reversible framework deformation of dynaCOF-301 undergoing deformation recovery and expansion during gas adsorption. We identified the moisture-induced rapid configuration exchange leading to a crystallinity enhancement upon water adsorption. We quantified the solvatochromic and hydrochromic effects as a guest-

adaptivity consequence of conformational/configurational changes coupled with electronic structural transition, which are of great interest in functional applications[43–48].

## Methods

### Materials

2,5-Dihydroxyterephthalaldehyde (dhTPA, purity > 98%) was purchased from TENSUS BIOTECH Co. Tetra-(4-anilyl)-methane (TAM, purity > 97%) was purchased from TENSUS BIOTECH Co and further purified according to the literature[49, 50]. Aniline (AR, ≥99.5%) was purchased from J & K Scientific Co. Glacial acetic acid (AR, ≥99.5%) was purchased from Sinopharm Chemical Reagent Co. Anhydrous 1,3,5-trimethylbenzen (AR, ≥ 98%) purchased from Energy Chemical Co. o-Dichlorobenzene (SP, ≥99%) was purchased from Aladdin Co.

### Synthesis of dynaCOF-301

A 20 ml vial was charged with tetra-(4-anilyl)-methane (TAM, 50 mg, 0.13 mmol), 2,5-dihydroxyterephthalaldehyde (dhTPA, 43.5 mg, 0.26 mmol), 5 ml anhydrous 1,3,5-trimethylbenzen and 5 ml o-dichlorobenzene. The mixture was dispersed under sonication; then, 0.15 ml aniline and 0.5 ml 6 M acetic acid were added to the vial. The reaction was heated for 3 days. The orange powder at the bottom of the vial was isolated by centrifugation and exhaustively washed by Soxhlet extractor with THF for at least 5 days. The sample was then transferred to a vacuum chamber, slowly heated to 100 °C to evacuate to 10 mTorr, and sustained for 24 h to give red powder of dynaCOF-301a. (40 mg, ca. 47.7% yield based on the dhTPA). Elemental analysis: Calcd. for $C_{41}H_{28}N_4O_4$: C, 76.86; H, 4.41; N, 8.74; O, 9.99%. Found: C, 74.65; H, 4.65; N, 9.0; O, 11.7%.

### Characterisation

The phase purity and crystallinity of samples were determined with a powder X-ray diffractometer (Bruker, D8 advance, Cu Kα). The crystal size and morphology were examined using a scanning electron microscope (JEOL, JSM 7800F Prime). The FT-IR spectra were collected on a PerkinElmer FT-IR Spectrometer equipped with ALPHA's Platinum ATR single reflection diamond ATR module. Thermogravimetric analyses (TGA) were performed on a TGA instrument (Perkin-Elmer, TGA 4000) with a heating rate of 5 °C min⁻¹ from ambient temperature to 800 °C under $N_2$ flow.

### Gas adsorption experiments

Gas adsorption measurements of $CO_2$ (195 K), $CH_4$ (112 K) and $N_2$ (77 K) were carried out on a 2-Ports microporosity and specific surface area analyser (Quantachrome, Autosorb IQ2). Static water adsorption measurements were performed on a precision vapour adsorption measuring system (BELSORP, MAXII). Dioxane vapour adsorption measurements were performed using a precision vapour adsorption system (BELSORP, Aqua3). DVS from Surface Measurement Systems (UK) is a gravimetric apparatus to measure the sample's mass change using continuous and constant nitrogen flow to bring the vapour to the sample.

### 3D electron diffraction analyses

A transmission electron microscope (JEOL F-200) with a hybrid pixel detector (ASI Cheetah1800) was used for 3D ED data collection. Low-dose electron beam, cryogenic sample holder (Gatan Model 914.6), and continuous tilting mode (-0.34°/s) were applied to achieve atomic-level resolutions for such beam-sensitive organic porous crystals. The activated sample was prepared by dispersed microcrystals in methanol and sonicated for 5 min. The suspension was dropped on a carbon film-supported TEM grid and transferred to the sample holder. The sample hold was inserted in the pre-vacuum chamber and vacuumed for

30 min. By accident, the atomic-level ED data for the solvated phase were collected during this process. The hydrated sample was prepared by pre-fumigation of the sample in water vapour for 12 h and dropped in liquid nitrogen for the cryogenic sample transfer process. Instamatic[51] was used for the data collection, and XDS programme package[52] was used for the data processing. Multiple datasets were merged for the hydrated phase to improve the data completeness. Crystal structure starting sets were solved through direct methods implemented in SIR2014 software[53], and the structure refinements were completed using SHELX[54] in Olex2 packages[55]. All non-hydrogen atoms were refined anisotropically, and the hydrogen was added geometrically and refined in a riding mode. The guest molecules in dynaCOF-301s could not be modelled, whose contribution to diffraction was deducted by solvent mask in the Olex2 package[55].

### Synchrotron PXRD analyses

The synchrotron PXRD data were collected at the Beamline BL14B1 of Shanghai Synchrotron Radiation Facility (SSRF) equipped with the capillary transmission mode with $\lambda = 0.6895$ Å. The activated sample of dynaCOF-301a was sealed under a high vacuum after heating at 120 °C in a borosilicate glass capillary with an outer diameter of 0.8 mm and a thickness of 0.01 mm. The sample of dynaCOF-301s was sealed under saturated THF vapour to avoid guest escape. The dynaCOF-301h was sealed at ambient under 60% humidity. Pawley refinement was first performed to refine the unit cell parameters of all three structures. Rietveld refinement was performed against synchrotron PXRD data with an initial model from single-crystal 3D ED structures. Four THF and four $H_2O$ molecules were added into all channels of dynaCOF-301s (THF) and dynaCOF-301h for further refinement. The dynaCOF-301-activated was refined with a structural model of dynaCOF-301h (structure solved against 3D ED data) without adding guest molecules.

### Solid-state NMR spectroscopy

All the ssNMR experiments were performed with magic angle spinning (MAS) on a Bruker AVANCE III HD 400 MHz wide-bore solid-state NMR spectrometer at a magnetic field of 9.4 T equipped with a standard Bruker MAS probe with a 3.2 mm (o.d.) zirconia rotor. ¹³C MAS NMR data were acquired at the Larmor frequency of 100.6 MHz. ¹³C chemical shifts were referenced to tetramethyl silane (TMS) at 0 ppm and calibrated using the carboxylic carbon of the glycine assigned to 176.2 ppm as a secondary reference. All the ¹³C experiments were carried out on a standard 3.2 mm double-resonance probe with a sample spinning rate of 12 kHz. ¹³C cross-polarisation (CP)-MAS experiments were carried out with a ¹H π/2 pulse length of 3.5 μs, a contact time of 2 ms, a pulse delay of 5 s, and a SPINAL-64 at a decoupling frequency of 81 kHz. The samples for ex-situ ssNMR experiments were prepared on a precision vapour adsorption system (BELSORP, Aqua3) by dosing dioxane vapour with varied pressures of 1,4-dioxane in the ssNMR rotor containing activated samples.

### In-situ PXRD measurements

A customised relative humidity or gas partial pressure controller was prepared (Supplementary Fig. 17). Two mass flow controllers (MFCs) with different controlling ranges (100 sccm and 5 sccm for low partial pressure of <5%) were connected to the $n$-$C_4H_{10}$ gas cylinder. Two MFCs were connected to the $N_2$ cylinder as a purge gas to activate the sample or control different relative humidity. The working gas was imported into the in-situ PXRD chamber (Bruker, D8 advance, Cu Kα). All the stainless-steel valves and joints were purchased from Shanghai X-tec Fluid Technology Co, Ltd, and the MFCs were purchased from Alicat Scientific (A Halma company).

### Band gap energy analyses

DRS spectra were recorded at room temperature on a Cary 5000 UV-vis spectrometer with an integrating sphere in the 200–800 nm wavelength range. The scan rate was

600 nm/min with a resolution of 1 nm $BaSO_4$ background was used as a reference. The band gap energies were derived from the *Tauc* plots[42] (Supplementary Figs. 21 and 23).

## Data availability

The data supporting this study's findings are available from the corresponding authors upon request. Crystallographic data for the structures reported in this Article have been deposited at the Cambridge Crystallographic Data Centre under deposition numbers CCDC 2238807 (dynaCOF-301s (ED)), 2238808 (dynaCOF-301h (ED)), 2238809 (dynaCOF-301s (Rietveld)), 2238810 (dynaCOF-301a (Rietveld)), and 2238811 (dynaCOF-301h (Rietveld)). Copies of the data can be obtained free of charge via https://www.ccdc.cam.ac.uk/structures/.

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

## Acknowledgements

This paper is dedicated to Prof. O. Terasaki on the occasion of his 80th birthday. The authors thank Prof. O. M. Yaghi and the Berkeley Global Science Institute for initiative support, Prof. W.-H. Zhu at ECUST and Dr. Z.-Y. Shen at ZhenGe BioTech for project support and Prof. Z. Li at ShanghaiTech University for beneficial discussion. We thank beamline BL14B at Shanghai Synchrotron Radiation Facility (SSRF) for providing beam time and assistance during data collection of high-resolution PXRD patterns, the Center of High-Resolution Electron Microscopy (ChEM, #EM02161943) for facility support, HPC Platform of ShanghaiTech University for MD simulation, Dr. N. Yu, Dr. M. Peng, and Ms. L. Long at the Analytical Instrumentation Center (SPST-AIC10112914) for technical support of PXRD, ssNMR and gas adsorption at ShanghaiTech University. This work is supported by the National Natural Science Foundation of China [Nos. 22271189 (Y.B.Z) and 22222108 (Y.M.)], the Science and Technology Commission of Shanghai Municipality [Nos. 21XD1402300 (Y.B.Z), 21JC1401700 (S.J. and Y.B.Z.), 21DZ2260400 (Y.B.Z, S.J., Y. M. and Y.Z.), and 22QC1401500 (Y.Z.)], and the Double First-Class Initiative Fund of ShanghaiTech University (Y.B.Z.).

## Author contributions

Y.B.Z. conceived and led the project. Y.X. synthesised and characterised the materials. T.S. performed the 3D cryo-ED analyses. T.Z. conducted the ssNMR spectroscopy analysis. X.Z. and S.J. conducted the MD simulation. S.L., X.Y. and Z.S. conducted the in-situ PXRD and DVS experiments. Y.X., T.Z. and W.W. collected the high-resolution PXRD data at SSRF. Y.M. conducted the Rietveld refinement of PXRD data. Y.X., T.Z., Y.Z. and Y.B.Z. wrote the manuscript and all the authors discussed and revised it together.

## Competing interests

The authors declare no competing interests.
