## [Peer Review File · Nature Communications]

REVIEWER COMMENTS

Reviewer #1 (Remarks to the Author):

This manuscript presents exciting findings on the topic of symmetry-breaking dynamics in a tautomeric 3D covalent organic framework (COF). Notably, the concept of enolimine-ketoenamine tautomerism, which has been successfully applied to 2D COFs (Ref. 15 and other references), is now well-established in 3D COFs. While the manuscript is well-written, I have some minor comments that should be addressed before acceptance.

Firstly, the authors did not discuss the chemical stability of the 3D COFs. It would be helpful to clarify whether these COFs are stable, as with the 2D beta-ketoenamine-based COFs. Secondly, the mechanism of host-guest interactions upon removal and adaptive inclusion of various guest molecules is unclear. To support this point, I suggest conducting an in-situ solid-state nuclear magnetic resonance (SSNMR) experiment.

Lastly, it is surprising that these 3D COFs are not porous. I had assumed that they would have higher porosity. It would be helpful to discuss the potential reasons for this unexpected observation.

Reviewer #2 (Remarks to the Author):

The authors have studied the structural adaptive nature of the 3D-imine linked covalent organic framework upon guest loading. Moreover, such structural change and interactions with guest molecules induce an iminol/cis-ketoenamine tautomerism which give rise to color change as a result of band-gap modification. The work reported by Zhang and col is well planned, executed and discussed and the results are relevant to being published in Nature Communications.

Authors' Responses to Reviewer Comments:

Reviewer #1:

This manuscript presents exciting findings on the topic of symmetry-breaking dynamics in a tautomeric 3D covalent organic framework (COF). Notably, the concept of enolimine-ketoenamine tautomerism, which has been successfully applied to 2D COFs (Ref. 15 and other references), is now well-established in 3D COFs. While the manuscript is well-written, I have some minor comments that should be addressed before acceptance.

Re: We are grateful for the reviewer's positive comments and constructive suggestions.

1. Firstly, the authors did not discuss the chemical stability of the 3D COFs. It would be helpful to clarify whether these COFs are stable, as with the 2D beta-ketoenamine-based COFs.

Re: We thank the reviewer for pointing out the important aspect of tautomerism. Yes, these COFs are more stable than the pristine COF-300. However, due to the partially enolimine-ketoenamine tautomerism, half of the linkages are still imine bonds. Thus, COF-301 is stable in basic but not in acidic conditions. According to the reviewer's suggestion, we have discussed this point in the revised manuscript:

“The chemical stability was highlighted by immersing the sample in 2 M NaOH, retaining its crystallinity for at least 7 days (Supplementary Fig. 2).”

2. Secondly, the mechanism of host-guest interactions upon removal and adaptive inclusion of various guest molecules is unclear. To support this point, I suggest conducting an in-situ solid-state nuclear magnetic resonance (ssNMR) experiment.

Re: We thank the constructive suggestion from the reviewer. To visualize the dynamic structure transformation of dynaCOF-301, we conducted the molecular dynamics simulation by compressing the expanded phase to the contracted phase. An animation has been provided as supplementary information. We have added discussion in the revised manuscript: “To gain energetic insight into the dynamic structural transformation, molecular dynamics (MD) simulations were implemented for statistical modelling of the transition from expanded to contracted phases of dynaCOF-301 (Supplementary Section 7). The internal pressure of dynaCOF-301 as a function of variable cell volumes was then calculated (Supplementary Fig. 32a). Stable structures with zero internal pressure are suggested with cell volumes of 3734 and 5844 Å³, comparable to observed cell volumes 3535 Å³ for dynaCOF-301a and 5570 Å³ for dynaCOF-301s, respectively. Surprisingly, a metastable structure with 5213 Å³ cell volume was suggested as an intermediate phase during the gas adsorption. From the energy profile (Supplementary Fig. 32b), the expanded framework is 60 kJ/mol higher energy than the contracted framework, whose energy barrier can be conquered by the adaptive inclusion of guest molecules.”

Per your suggestion, we have conducted the ex-situ ssNMR spectroscopy by dosing various amounts of 1,4-dioxane vapour into the sample. We have added the discussion in the revised manuscript and the data in the revised supplementary information: “Temperature sensitivity of the enolimine-ketoenamine tautomerism was further confirmed by variable-temperature ssNMR for dynaCOF-301a showing distinguishable intensity changes of chemical shifts of C9 and C9' at 243-352 K (Supplementary Fig. 30). We also conducted the ex-situ ssNMR experiments by dosing various amounts 1,4-dioxane vapour to dynaCOF-301a for probing the local chemical environment changes of host framework upon adaptive inclusion of guest molecules (Supplementary Fig. 31). Upon 1 mol mol⁻¹ inclusion of 1,4-dioxane, the chemical shifts of C9 and C9' converged to one chemical shift at 153.0 ppm implying the tautomerism to diiminol form and the gradual recovery of high symmetry. Further inclusion of 1,4-dioxane leads to the chemical shifts of C6 and C6' converging to one chemical shift at 160.7 ppm, indicating the conformation change of imine bonds.”

Supplementary Figure 30. ^{13}C SSNMR of dynaCOF-301a in varied temperatures, showing a distinguishable effect on the relative intensity of C9 and C9', implying temperature-dependent tautomerism of moieties.

Supplementary Figure 31. ex-situ ^{13}C SSNMR of dynaCOF-301 loading with different pressures of dioxane vapour, showing a distinguishable effect on the chemical shift, implying guest-dependent tautomerism of moieties.

3. Lastly, it is surprising that these 3D COFs are not porous. I had assumed that they would have higher porosity. It would be helpful to discuss the potential reasons for this unexpected observation.

Re: Although these COFs are not highly porous due to the interwoven network and crystal contraction upon activation, we have to clarify that the COF-301 is porous through the adaptive inclusion of guest molecules, which have shown stepwise adsorption isotherms for N_2 (77 K, $350 \text{ cm}^3 \text{ g}^{-1}$) and a pore volume of $0.57 \text{ cm}^3 \text{ g}^{-1}$. We have discussed this in the revised manuscript: A significantly higher N_2 uptake ($350 \text{ cm}^3 \text{ g}^{-1}$) and a larger pore volume ($0.57 \text{ cm}^3 \text{ g}^{-1}$) were observed for dynaCOF-301 (Fig. 2f) than those of the dynaCOF-300 (only $\sim 200 \text{ cm}^3 \text{ g}^{-1}$ and $0.32 \text{ cm}^3 \text{ g}^{-1}$), suggesting a full crystal expansion for dynaCOF-301 but only partial conversion for dynaCOF-300.

Reviewer #2:

The authors have studied the structural adaptative nature of the 3D-imine linked covalent organic framework upon guest loading. Moreover, such structural change and interactions with guest molecules induce an iminol/*cis*-ketoenamine tautomerism which give rise to color sample as a result of from band-gap modification. The work reported by Zhang and coworkers is well planned, executed and discussed and the results are relevant to being published in Nature Communications.

Re: We thank the reviewer for the insightful comments and positive evaluation. We appreciate the reviewer for pointing out the novelty of this work. Indeed, this work represents an early example in 3D COFs with guest-induced crystal structural transformation to control iminol/*cis*-ketoenamine tautomerism accompanied with colour change and band-gap modification.

REVIEWERS' COMMENTS

Reviewer #1 (Remarks to the Author):

This manuscript can be accepted now.